# Spatial Survival Model for COVID-19 in México

**DOI:** 10.3390/healthcare12030306

**Published:** 2024-01-24

**Authors:** Eduardo Pérez-Castro, María Guzmán-Martínez, Flaviano Godínez-Jaimes, Ramón Reyes-Carreto, Cruz Vargas-de-León, Alejandro Iván Aguirre-Salado

**Affiliations:** 1Unidad de Investigación de Salud en el Trabajo, Centro Médico Nacional Siglo XXI, Ciudad de México 06720, Mexico; 14579645@uagro.mx; 2Facultad de Matemáticas, Universidad Autónoma de Guerrero, Chilpancingo 39087, Mexico; fgodinezj@uagro.mx (F.G.-J.); rcarreto@uagro.mx (R.R.-C.); 3Sección de Estudios de Posgrado, Escuela Superior de Medicina, Instituto Politécnico Nacional, Ciudad de México 11340, Mexico; 4División de Investigación, Hospital Juárez de México, Ciudad de México 07760, Mexico; 5Instituto de Física y Matemáticas, Universidad Tecnológica de la Mixteca, Huajuapan de León 69000, Mexico; aleaguirre@mixteco.utm.mx

**Keywords:** bayesian methodology, proportional hazard frailty model, spatial correlation, survival time

## Abstract

A spatial survival analysis was performed to identify some of the factors that influence the survival of patients with COVID-19 in the states of Guerrero, México, and Chihuahua. The data that we analyzed correspond to the period from 28 February 2020 to 24 November 2021. A Cox proportional hazards frailty model and a Cox proportional hazards model were fitted. For both models, the estimation of the parameters was carried out using the Bayesian approach. According to the DIC, WAIC, and LPML criteria, the spatial model was better. The analysis showed that the spatial effect influences the survival times of patients with COVID-19. The spatial survival analysis also revealed that age, gender, and the presence of comorbidities, which vary between states, and the development of pneumonia increase the risk of death from COVID-19.

## 1. Introduction

The coronavirus disease COVID-19, which is caused by the SARS-CoV-2 virus (Severe Acute Respiratory Syndrome Coronavirus type 2), was first reported in Wuhan City, China, on 31 December 2019, and on 11 March 2020, COVID-19 was declared a pandemic by the World Health Organization (WHO) [1]. The first person with COVID-19 in México was detected on February 27 of the same year [2].

México is a country with a high prevalence of comorbidities, such as hypertension, obesity, and diabetes. Given that it has been shown that the existence of comorbidities associated with SARS-CoV-2 infection increases the risk of mortality [3], México faced a severe crisis during the COVID-19 pandemic.

The risk factors of SARS-CoV-2 have been investigated using the survival analysis methodology [4]. In México, previous studies have investigated the risk factors associated with deaths due to COVID-19 using survival analysis. It was reported that male gender; advanced age; the presence of comorbidities such as chronic kidney disease, diabetes, and arterial hypertension; the development of pneumonia; hospitalization; admission to an intensive care unit (ICU); intubation; geographic location; and the health sector where the patient was treated are associated with lower survival in patients affected by SARS-CoV-2 [5,6,7]. Furthermore, [8] found that poorer population groups have lower COVID-19 survival rates.

Survival data are analyzed using proportional hazards regressions [9,10]. Considering that data can come from different locations, regions, or zones and that the number of cases varies between locations, it is appropriate to include location information in a survival model. The variability of data in space can be included in a survival model by means of a random effect or term [11].

Statistical models that take geographic location into account are increasingly used in survival analysis [12]. This is because they (i) incorporate spatial variation in survival times [13]; (ii) implement the Bayesian approach for parameter estimation and the new tools of Geographic Information Systems [13,14,15]; (iii) surpass the classical models by reducing the bias in the estimates and incorporating geographic information of the data [16,17]; (iv) introduce information about the spatial locations of the data, which plays an important role in predicting survival as it serves as a proxy for unmeasured regional characteristics such as access to medical care [13,18]; and (v) provide relevant information for public health decision making, such as the extent and direction of disease spread or the locations of disease hotspots, allowing control measures to be more effective and facilitating the adequate allocation of health resources [19].

Some papers using survival analysis in the health field that take the spatial information of data into account include Bayesian spatial survival modeling for dengue fever in Indonesia [20]; modeling the time to detection of urban tuberculosis in Portugal [21]; modeling of spatially correlated survival data for people with different types of cancer [22]; modeling of spatial variation in leukemia survival data [23]; parametric normal transformation models for spatially correlated, right-censored survival data [24]; Bayesian Weibull and Cox semi-parametric spatial models to describe a data set on dengue hospitalization [25]; an estimate of recovery times of COVID-19 patients in India [26]; a hierarchical conditional autoregressive model for colorectal cancer survival data [27]; a semi-parametric model with the cure fraction for multivariate time-to-event data [28]; a proportional hazards spatial frailty model to determine risk factors associated with under-five mortality in Kenya [29]; a hierarchical Bayesian model to jointly model longitudinal and survival data considering the effects of spatial and temporal frailty for AIDS data [30]; and a proportional hazards models with marginal cure rates to identify risk factors for tooth loss and predict the remaining useful life of a patient’s teeth [31].

Previous studies conducted in México did not consider the spatial effect on the survival of patients with COVID-19. Therefore, this paper aims to investigate the risk factors associated with COVID-19 deaths in three Mexican states, considering the spatial variability of the data in the statistical model.

## 2. Study Area

México is a country located in North America; it has 32 Federal Entities, also known as states. Each state is made up of a certain number of municipalities; there are a total of 2475 municipalities throughout the country. The state with the most municipalities is Oaxaca with 570, followed by Puebla with 217, Veracruz with 212, and Jalisco and the State of México with 125 municipalities each. The states with the fewest municipalities are Baja California with five and Baja California Sur with five (Figure 1). In the year 2020, México’s total population was 126,014,024 people [32]. The states with the highest populations were the State of México with 16,992,418 people, México City with 9,209,944 people, Veracruz with 8,062,579 people, Jalisco with 8,348,151 people, and Puebla with 6,583,278 people, while the states with the lowest populations were Colima with 731,391 people and Baja California Sur with 798,447 people (Figure 1).

According to the 2020 Population and Housing Census conducted by the National Institute of Statistics and Geography (INEGI), the population density in México was 64 inhabitants per square kilometer (hab/km^2^) [32]. The INEGI categorizes the 32 Federal Entities as high, medium, or low population density (Figure 1).

The states with high population densities are Colima, Aguascalientes, Guanajuato, Querétaro, Hidalgo, the State of México, Morelos, México City, Puebla, and Tlaxcala; the states with medium population densities are Baja California, Sinaloa, Nuevo León, Jalisco, Michoacán de Ocampo, Guerrero, Veracruz de Ignacio de la Llave (Veracruz), Tabasco, Chiapas, and Yucatán; and the states with low population densities are Baja California Sur, Campeche, Chihuahua, Coahuila, Durango, Nayarit, Oaxaca, Quintana Roo, San Luis Potosí, Sonora, Tamaulipas, and Zacatecas (Figure 1).

To analyze the survival times of people infected with SARS-CoV-2 in space, we worked with the State of México (red polygon in Figure 2) with high population density, 125 municipalities, and a population of 16,992,418; Guerrero (orange polygon in Figure 2) with medium population density, 81 municipalities, and a population of 3,540,685; and Chihuahua (yellow polygon in Figure 2), with low population density, 67 municipalities, and a population of 3,741,869.

Statistical analyses were performed in R software version 4.1.2 [33] (The R Foundation for Statistical Computing, Vienna, Austria).

### 2.1. Database

This study used the open-access database of suspected cases of COVID-19 published by the Ministry of Health of México through the Epidemiological Surveillance System for Respiratory Viral Diseases [34]. In México, as of 24 November 2021, a cumulative total of 3,872,263 cases had been confirmed. However, the open-access database contains 2,028,000 records of confirmed COVID-19 patients reported by the laboratory of the National Network of Epidemiological Surveillance Laboratories and private laboratories endorsed by the Institute of Epidemiological Diagnosis and Reference [33]. The analyzed data correspond to the period from 28 February 2020, to 24 November 2021.

The variables considered in this study were age, gender, pneumonia, diabetes, chronic obstructive pulmonary disease (COPD), cardiovascular disease (CVD), obesity, asthma, chronic kidney disease (CKD), and hypertension (Table 1).

The patients’ sociodemographic variables were not considered because the Ministry of Health, which collected the information, did not include them and there was no way to obtain them for the patients considered in this study.

The response variable was survival time, which was defined as the time between the date of symptom onset and patient death. The data were censored on 24 November 2021, for people who were alive at the end of the study period.

### 2.2. Spatial Autocorrelation

To study the survival times in the three states of México, we first determined, using the Moran index, the degree of spatial association that existed in the confirmed cases of COVID-19. Moran’s index ($I_{G}$) is defined as
(1)$$I_{G}=\frac{m\sum_{i}^{m}\sum_{j}^{m}w_{ij}\left(Y_{i}-\bar{Y}\right)\left(Y_{j}-\bar{Y}\right)}{\sum_{i\neq j}^{m}w_{ij}\sum_{i}^{m}{\left(Y_{i}-\bar{Y}\right)}^{2}}$$
where $Y_{i}$ and $Y_{j}$ are the values of the variable Y in localities i and j, respectively; $\bar{Y}$ is the average of the variable Y in the m localities under study; and $w_{ij}$ indicates the proximity between units i and j. When locations i and j are neighbors, $w_{ij}=1$. Otherwise, $w_{ij}=0$. For i=j, $w_{ij}=1$. The hypotheses set for $I_{G}$ are given as follows: $H_{0}$: the spatial pattern of Y is random vs. $H_{1}$: the spatial pattern of Y is not random. $H_{0}$ is rejected if *p*-value < α.

In this work, the localities are the municipalities of each of the three states. Variable Y is the number of confirmed cases of COVID-19. Two municipalities are neighbors if they share at least one point on the border, which can be a vertex and/or at least one border (Queen). Thus, the distance function is given by an indicator function, where $w_{ij}=1$ if the municipalities share a vertex and/or at least one border. Also, $w_{ij}=1$ if i=j. Otherwise, $w_{ij}=0$ and i,j=1,…,m. With the $w_{ij}$ values, we constructed the weight matrix (W) of dimension m×m, assuming that $\sum_{i=1}^{m}\sum_{j=1}^{m}=m$. For the State of México, m=125; for Guerrero, m=81; and for Chihuahua, m=67. The calculation of the Moran index was performed with the *moran.mc* function of the *spded* package in R statistical software version 4.1.2 (The R Foundation for Statistical Computing, Vienna, Austria) [33].

### 2.3. Statistical Model

Very often, time-to-event data are grouped into strata (clusters), such as clinical sites, geographic regions, and so on. $t_{ij}$ is the time to death or censoring for the j-th subject at location $s_{i}$; $i=1,\ldots ,m;\;j=1,\ldots ,n_{i}$; and the total number of subjects under study is $n=\sum_{i=1}^{m}n_{i}$. $x_{ij}$ is a vector of covariates of dimension p, and $\beta ={\left(\beta_{1},\ldots ,\beta_{p}\right)}^{\prime }$ is a vector of regression coefficients. The proportional hazard (PH) model (Model (2)) is shown below:(2)$$h\left(t_{ij}\right)=h_{0}\left(t_{ij}\right){exp}^{\beta^{\prime }x_{ij}}$$
where $h_{0}$ is the baseline hazard.

The proportional hazard frailty model (Model (3)) is shown below:(3)$$h\left(t_{ij}\right)=h_{0}\left(t_{ij}\right)e^{\beta^{\prime }x_{ij}+v_{i}}$$
where $v_{i}$ is an unobserved frailty associated with $s_{i}$ and is designed to capture differences among the strata. In the spatial survival analysis, subjects are in m distinct regions ($s_{1},\ldots ,s_{m}$).

The PH frailty model has a survival function:(4)$$S\left(t_{ij}\right)=S_{0}{\left(t_{ij}\right)}^{e^{\beta^{\prime }x_{ij}+v_{i}}}$$
with a density function:(5)$$f_{x}\left(t_{ij}\right)=e^{\beta^{\prime }x_{ij}+v_{i}}S_{0}{\left(t_{ij}\right)}^{e^{\beta^{\prime }x_{ij}+v_{i}}-1}f_{0}\left(t_{ij}\right)$$
where $S_{0}\left(\cdot \right)$ is the baseline survival function and $f_{0}\left(\cdot \right)$ is the baseline density function, which are assumed to be unique for all individuals. For this study, $t_{ij}$ values are right-censored $\left(a_{ij},\infty \right)$. $t_{ij}$ is a survival time if $\delta_{ij}=1$, and it is a censoring time if $\delta_{ij}=0$ [35]. The likelihood of a PH frailty model is given by
(6)$$L=\prod_{i}^{m}\prod_{j}^{n_{i}}{\left[h\left(t_{ij}\right)\right]}^{\delta_{ij}}S\left(t_{ij}\right)$$

#### 2.3.1. Prior Distributions

The parameters of the PH model have the following prior distributions:(7)$$\beta ∼N_{p}\left(\beta_{0},S_{0}\right)$$
(8)$$S_{0}\left(\cdot \right)|\alpha ,\theta ∼{TBP}_{L}\left(\alpha ,S_{\theta }\left(\cdot \right)\right),\alpha ∼\Gamma \left(a_{0},b_{0}\right),\theta ∼N_{2}\left(\theta_{0},V_{0}\right)$$
(9)$${\left(v_{1},\ldots ,v_{m}\right)}^{\prime }|\tau ,\phi ∼GRF\left(\tau^{2},\phi \right),\tau^{-2}∼\Gamma \left(a_{\tau },b_{\tau }\right),\phi ∼\Gamma \left(a_{\phi },b_{\phi }\right)$$
where ${TBP}_{L}$ refers to the transformed Bernstein polynomial [35]. For a fixed positive integer (L), the prior ${TBP}_{L}\left(\alpha ,S_{\theta }\left(\cdot \right)\right)$ is defined as
(10)$$S_{0}\left(t\right)=\sum_{j=1}^{L}w_{j}I\left(S_{\theta }\left(t\right)|j,L-j+1\right),w_{L}∼Dirichlet\left(\alpha ,\ldots ,\alpha \right)$$
where $w_{L}={\left(w_{1},\ldots ,w_{L}\right)}^{\prime }$ is a vector of positive weights, $I\left(\cdot |a,b\right)$ is a beta cumulative distribution function with parameters $\left(a,b\right)$, and $\left\{S_{\theta }\left(\cdot \right):\theta \in \Theta \right\}$ is a parametric family of survival functions with support on positive reals. Furthermore, a log-logistic distribution for $S_{\theta }\left(t\right)$ is assumed; that is,
(11)$$S_{\theta }\left(t\right)={\left\{1+{\left(e^{\theta_{1}}t\right)}^{exp\left(\theta_{2}\right)}\right\}}^{-1}$$
where $\theta ={\left(\theta_{1},\theta_{2}\right)}^{\prime }$.

For $v_{i}=v\left(s_{i}\right)$, a Gaussian random field (GRF) $\left\{v\left(s\right),s\in S\right\}$ is assumed, where $v={\left(v_{1},\ldots ,v_{m}\right)}^{\prime }$ follows a multivariate Gaussian distribution, $v∼N_{m}\left(0,\tau^{2}R\right)$, $\tau^{2}$ measures the amount of spatial variation across locations, and the $\left(i,j\right)$ element of R is modeled as
(12)$$R\left[i,j\right]=\rho \left(s_{i},s_{j};\phi \right)=exp\left\{-{\left(\phi ∥s_{i}-s_{j}∥\right)}^{k}\right\}$$
which is a correlation function controlling the spatial dependence of $v\left(s\right)$. ϕ>0 is a range parameter controlling the spatial decay over distance, k∈(0, 2] is a shape parameter, and $∥s_{i}-s_{j}∥$ is the distance between $s_{i}$ and $s_{j}$. The prior $GRF\left(\tau^{2},\phi \right)$ is defined as
(13)$$v_{i}|{\left\{v_{j}\right\}}_{j\neq i}∼N\left(-\sum_{\left\{j:j\neq i\right\}}r_{ij}v_{j}/r_{ii},\;\;\;\tau^{2}/r_{ii}\right)\;\;\;\;\;i=1,\ldots ,\;m$$
where $r_{ij}$ is the $\left(i,j\right)$ element of $R^{-1}$ [35].

The spatial dependence of COVID-19 survival times is captured in the covariance structure (R) of the Gaussian random field v, which is assumed to be a stationary process.

#### 2.3.2. Posterior Distributions

For Equation (2), the likelihood function is
(14)$$L_{Cox}=\prod_{i}^{m}\prod_{j}^{n_{i}}{\left[h_{0}\left(t_{ij}\right)e^{\beta^{\prime }x_{ij}}\right]}^{\delta_{ij}}{S_{0}\left(t_{ij}\right)}^{e^{\beta^{\prime }x_{ij}}}$$

Therefore, the posterior distribution is
(15)$$\pi_{Cox}\propto L_{Cox}\pi \left(w_{J}|\alpha \right)\pi \left(\alpha \right)\pi \left(\beta \right)\pi \left(\theta \right)$$

For Equation (3), the likelihood function is
(16)$$L_{Spatial}=\prod_{i}^{m}\prod_{j}^{n_{i}}{\left[h_{0}\left(t_{ij}\right)e^{\beta^{\prime }x_{ij}+v_{i}}\right]}^{\delta_{ij}}{S_{0}\left(t_{ij}\right)}^{e^{\beta^{\prime }x_{ij}+v_{i}}}$$

Therefore, the posterior distribution is
(17)$$\pi_{Spatial}\propto L_{Spatial}\pi \left(w_{J}|\alpha \right)\pi \left(\alpha \right)\pi \left(\beta \right)\pi \left(\theta \right)\pi \left(\tau^{-2}\right)\pi \left(\phi \right)$$

A Bayesian fitting of the proportional hazard frailty model was obtained using R software and several libraries, such as *spBayesSurv* [36]. The function *survregbayes* set the following hyperparameters as defaults: $\beta_{0}=0$, $S_{0}=10^{10}I_{p}$, $\theta_{0}=\hat{\theta }$, $V_{0}=10\hat{V}$, $a_{0}=1$, $b_{0}=1$, $a_{\tau }=0.001$, and $b_{\tau }=0.001$, where $\hat{\theta }$ is the maximum likelihood estimate of θ and $Cov\left(\theta \right)=\hat{V}$.

To analyze the survival times of people with COVID-19, the proportional hazard frailty model was used (Model (3)). Right censoring was used, and the model’s covariates were age, gender, pneumonia, diabetes, chronic obstructive pulmonary disease, cardiovascular disease, obesity, asthma, chronic kidney disease, and hypertension. For Guerrero, the analysis was carried out with the 37,278 registered cases of COVID-19; for Chihuahua, we worked with the 45,954 observed cases of COVID-19; and for the State of México, a sample of 100,000 observations out of the 176,268 available observations were used. The initial values of the hyperparameters of the prior distributions (Equations (5) and (6)) were set at $a_{0}=4$ and $b_{0}=4$ for Chihuahua and the State of México and at $a_{0}=1$ and $b_{0}=1$ for Guerrero. For the three states, the hyperparameters were fixed at L=15, $a_{\tau }=5$, $b_{\tau }=5$, $a_{\phi }=6$, and $b_{\phi }=3$. For the estimation of the parameters of Model 3, Markov Chain Monte Carlo (MCMC) algorithms were used, which consisted of two chains of 25,000 iterations, with a thinning of 10 and a burning of 5000 samples.

The MCMC procedure is carried out with the Gelman and Rubin convergence diagnostic ($\hat{R}$) [37,38], which is implemented in the library *coda* [39]. Values of $\hat{R}$ close to one give evidence of the convergence of the MCMC [37]. Another diagnostic is the MCMC traceplot (plot of the values in the simulated chains vs. the iteration). Good mixing of the chains indicates convergence of the MCMC. Cox–Snell residuals [35] can be used to verify the proportional hazards assumption [40]. However, testing this assumption numerically or graphically is complicated since the proportional hazards hypothesis only approximates the correct model for one covariate and any formal test based on a sufficiently large sample will reject the null hypothesis of proportionality [41]. In any case, Cox–Snell residuals should be considered to determine the model fit [42].

A classical survival analysis was also performed using the Cox model (Model (2)). For the three states, $a_{0}=6$, $b_{0}=6$, L=15, $a_{\tau }=5$, $b_{\tau }=5$, $a_{\phi }=6$, and $b_{\phi }=3$ were considered. However, only the results of the best model are reported, which for the three states, was the proportional hazard frailty model. We used three criteria to compare the fitted models: the deviance information criteria (DIC) [43], the Watanabe–Akaike information criterion (WAIC) [44], and the log pseudo marginal likelihood (LPLM) [45]. Generally, smaller DIC and WAIC values show good model fitting, while larger LPML values indicate a better predictive performance of a model.

The figures were created using the INEGI’s shapefiles (https://www.inegi.org.mx/app/biblioteca/ficha.html?upc=889463807469 (accessed on 26 November 2021)), which are freely available for academic use and other non-commercial uses [46].

## 3. Results

### 3.1. State of México

As of 24 November 2021, 176,268 confirmed cases of COVID-19 were registered in the State of México. The municipalities with more than 10,000 cases were located in the east and center of the state: Ecatepec de Morelos (22,613 cases), Nezahualcóyotl (16,992 cases), Toluca (13,647 cases), Naucalpan de Juárez (12,450), and Tlalneplantla de Baz (10,605 cases). The municipalities with less than 20 cases were located in the southeast of the state, including Zacazonapan (18 cases), Ixtapan del Oro (14 cases), and Otzoloapan (12 cases) (Figure 3).

According to the Moran index, in the State of México, confirmed COVID-19 cases were more correlated in the municipalities that are neighbors than in the municipalities that are not ($I_{G}=0.363,\;p-value<0.05$). Therefore, it was convenient to use the proportional hazard frailty model to model COVID-19 survival times.

The estimations of the posterior means and medians of the parameters, as well as their 95% credible intervals (CrIs), and the Gelman and Rubin convergence diagnostic values for the spatial model (proportional hazard frailty model) of the State of México are presented in Table 2. Because the values of $\hat{R}$ are close to 1, there is convergence in the model chains [37]. The chains show good mixing in Figure A1 (Appendix A). However, there is no indication that the Cox assumption is violated since no large deviations from the Cox–Snell residuals are observed in Figure A2 (Appendix A).

Male gender and age were factors that increased the risk of death from COVID-19, while the COPD variable was not significant. According to Table 2 and taking into account the hazard ratios (HRs) of the regression coefficients, CKD (HR = 1.588, 95% CrI: 1.482–1.707), diabetes (HR = 1.178, 95% CrI: 1.135–1.217), obesity (HR = 1.137, 95% CrI: 1.088–1.1852), and hypertension (HR = 1.065, 95% CrI: 1.026–1.102) were the comorbidities identified as risk factors. People who developed pneumonia had an increased risk of death (HR = 5.365, 95% CrI: 5.181–5.578). The estimated risk of death was 20% lower for people with asthma (HR = 0.796, 95% CrI: 0.677–0.922).

The estimates of $TBP_{L}$ and GRF were also significant. The variance of the spatial term v, which was given by ${\hat{\tau }}^{2}=1.540$, was significant, with a 95% CrI of 0.936–2.668 (Table 2). Therefore, the inclusion of the random effect was relevant, which means that the risk of death from COVID-19 was not homogeneous across the 125 municipalities of the State of México.

Figure 4 shows the average values of posterior sampling for frailties ($v_{i}$). We can identify clusters of municipalities. The inhabitants of the eastern municipalities had the highest risk of mortality adjusted for the effects of covariates.

### 3.2. State of Guerrero

As of 24 November 2021, 37,278 confirmed cases of COVID-19 were registered in Guerrero. The municipalities with more than 1000 cases were located in the south, center, and north of the state: Acapulco de Juárez (14,639 cases), Chilpancingo de los Bravo (6773 cases), Zihuatanejo de Azueta (1984 cases), Iguala de la Independencia (1664 cases), and Taxco de Alarcon (1007 cases). The municipalities with the fewest cases were located in the east and north of the state: General Canuto A. Neri (nine cases), Atlamajalcingo del Monte (eight cases), Tlacoapa (six cases), Pedro Ascencio Alquisiras (four cases), and Iliatenco (four cases) (Figure 5).

According to the Moran index, in the state of Guerrero, the confirmed COVID-19 cases were more correlated in the municipalities that are neighbors than in the municipalities that are not ($I_{G}=0.135,\;p-value<0.05$).

Estimates of the means and posterior medians of the parameters, their corresponding 95% CrIs, and the values of $\hat{R}$ (which indicate convergence) are presented in Table 3. Figure A3 (Appendix A) reveals that the MCMC chains of the estimated parameters show good mixing. On the other hand, there is no indication that the Cox assumption is violated since no large deviations from the Cox–Snell residuals are observed in Figure A4 (Appendix A).

Asthma was not significant, while male gender and age were factors that increased the risk of death in patients with COVID-19.

Based on Table 3, the comorbidities associated with an increased risk of death from COVID-19 were diabetes (HR = 1.355, 95% CrI: 1.270–1.444), obesity (HR = 1.243, 95% CrI: 1.157–1.344), CVD (HR = 1.228, 95% CrI: 1.065–1.424), CKD (HR = 1.217, 95% CrI: 1.075–1.384), COPD (HR = 1.150, 95% CrI: 1.001–1.320), and hypertension (HR = 1.073, 95% CrI: 1.004–1.141), as well as the development of pneumonia (HR = 16.776, 95% CrI: 15.150–18.411).

The estimates of $TBP_{L}$ and GRF were also significant. The posterior mean of the variance (${\hat{\tau }}^{2}=1.753$) of the spatial PH model was significant, with a 95% CrI of 1.015–3.255 (Table 3). Therefore, the random effects had an influence. This means that the risk of COVID-19 was not homogeneous across the 81 municipalities of the state of Guerrero.

Figure 6 shows the quantiles of the average values of the posterior samples of the spatial term v. We can identify clusters of municipalities. People with COVID-19 in the southern municipalities of Guerrero and, in general, those located on the periphery of the state presented the highest mortality risk adjusted for covariate effects.

### 3.3. State of Chihuahua

As of 24 November 2021, 45,954 confirmed COVID-19 cases were registered in the state of Chihuahua. The municipalities with more than 1000 cases were located in the central and eastern parts of the state: Chihuahua (12,582 cases), Delicias (1842 cases), Hidalgo del Parral (1750 cases), and Cuauhtémoc (1687 cases). The municipalities with less than five cases were located in the west of the state: Matachí (three cases), Huejotitlán (two cases), and Maguarichi (one case) (Figure 7).

Although the confirmed cases of COVID-19 in the state of Chihuahua presented a random pattern ($I_{G}=-0.0423,p-value=0.72$), we used the proportional hazard frailty model to study the COVID-19 survival times.

Table 4 presents the means and medians of the estimated parameters with their respective 95% CrIs and $\hat{R}$ values. The Gelman and Rubin statistic ($\hat{R}$) has values close to 1, which implies that there is convergence in the Markov chains. Figure A5 (Appendix A) shows a good mix of the chains of the parameters of the PH frailty model. On the other hand, there is no indication that the Cox assumption is violated since no large deviations from the Cox–Snell residuals are observed in Figure A6 (Appendix A).

As in the State of México and Guerrero, male gender and age were associated with a higher risk of death. The variables that were not significant were COPD, CVD, and asthma.

According to Table 4, the comorbidities associated with a higher risk of death were CKD (HR = 1.704, 95% CrI: 1.521–1.883), diabetes (HR = 1.406, 95% CrI: 1.327–1.493), obesity (HR = 1.320, 95% CrI: 1.235–1.407), hypertension (HR = 1.277, 95% CrI: 1.200–1.351), and the development of pneumonia (HR = 5.408, 95% CrI: 5.103–5.754).

The variance of v (${\hat{\tau }}^{2}=2.175$) was significant, with a 95% CrI of 1.221–3.706 (Table 4). The random effect had an influence, which means that the risk of COVID-19 was not homogeneous across the 67 municipalities of the state of Chihuahua. In general, the municipalities with the most cases of COVID-19 presented larger values of v. This was the case for the municipalities in the north and southeast of the state of Chihuahua, which had the highest risk of mortality adjusted for the effects of covariates (Figure 8).

Finally, according to the DIC and WAIC, models that incorporate the spatial effect have better fits than models that do not (Table 5).

For the LPML, the three spatial models also have better predictive fits than the nonspatial models. Therefore, the relevance of using the proportional hazard frailty models to model COVID-19 survival times is verified.

## 4. Discussion

In this study, we present a different approach to modeling the spatial dependence of the survival times of patients with COVID-19 in three Mexican states. Spatial heterogeneity, sociodemographic variables, and individual characteristics, such as comorbidities, were found to be associated with the severity of COVID-19, as reported in several investigations.

According to the DIC and WAIC, models incorporating a spatial term have a better predictive fit than Cox models without a frailty term. This is because classical survival models cannot account for the spatial correlation of the data and, therefore, should not be considered as the standard model in research when geographic information is available [12,16]. These results are in agreement with those reported by Thamrin et al. [20], who compared a Bayesian spatial survival model with a nonspatial one to analyze the factors influencing the survival of dengue patients and found that the spatial model was more appropriate. Daniel et al. [29] found that a spatial Cox proportional hazards model performed better compared with a nonspatial model in identifying risk factors for under-five mortality. In a study by Mahanta et al. [26] related to recovery times in patients with COVID-19, a spatial survival model presented a better fit than a model without a frailty term.

According to the obtained results, men have a higher risk of dying than women, age is associated with a higher risk, and certain comorbidities can increase this risk, which is consistent with several studies reported in the literature [5,6,7,8,47].

Diabetes, hypertension, obesity, and CKD were the only statistically significant comorbidities in the three spatial survival models. The impact of comorbidities on the risk of death from COVID-19 is well known; however, the Mexican population has a high prevalence of metabolic diseases, which makes it vulnerable to developing complications from COVID-19. Mexico is second worldwide in the prevalence of obesity; according to the ENSANUT 2018 survey, 75.2% of the Mexican population over 20 years of age is overweight or obese. In addition, the prevalence of diabetes in Mexicans over 20 years of age is 10.3%, and the prevalence of hypertension is 18.4% in patients over 20 years of age [48]. For Chihuahua and the State of Mexico, CKD was associated with a higher probability of dying. According to other studies on comorbidities recorded in Mexican data sets, CKD posed the highest risk of severe COVID-19 in Mexico [49]. In contrast, in the state of Guerrero, the highest risk was posed by diabetes.

It is worth mentioning that for the State of México, people who had asthma had a lower risk of dying compared with people who did not have asthma. Previous studies have reported this “protective” effect [50,51]. Some authors report that asthma may protect against the fatal outcomes of COVID-19 due to several possible mechanisms, such as the use of inhaled corticosteroids, chronic inflammation, reduced viral exposure due to protection, and/or mucus hypersecretion [51,52].

People with pneumonia in the state of Guerrero had a higher risk of dying compared with people in the states of Chihuahua and México (HR = 16.776, 95% CrI: 15.150–18.411). A probable cause of this high risk of death is that Guerrero is one of the states with the worst health services in México, both in the perception of its population and due to a lack of supplies and medicines, and the lack of medical coverage is particularly severe in rural areas, which could have led to poor care for people who developed severe pneumonia. In fact, living in the southern region of the country is related to the severity of COVID-19. These disparities based on geographic location and ethnicity are closely linked to socioeconomic inequality: the southern region has higher rates of poverty and a concentration of indigenous people, which shows how different forms of inequality intersect [53].

There is evidence of geographic disparities in COVID-19 survival times in the three states that were analyzed, which may be influenced by socioeconomic factors, demographic factors, and health-related lifestyle factors. The impacts of these comorbidities vary by geographic location, and some are more important predictors of the risk of COVID-19 death in some states than in others. This demonstrates the importance of using global and local models, in this case, at the state level, to investigate the determinants of geographic disparities in health outcomes and health services utilization [54,55].

To our knowledge, this is the first study in México to use the spatial survival analysis approach to study risk factors associated with COVID-19 deaths that exhibit spatial variability across these three states.

The limitations of this work are as follows: First, the data set did not include clinical variables related to the evolution of the diseases due to COVID-19, which could have been useful for adjusting the model. Secondly, the variables that allowed us to identify the presence of comorbidities were self-reported; therefore, misclassification bias is very likely. Thirdly, the sociodemographic variables of the patients, such as age and income deciles, or access to basic services, such as drinking water and drainage, among others, were not recorded in the database. In future work, we intend to consider information related to socioeconomic aspects at the municipal level, such as the level of social progress, the level of urbanization, the level of poverty, and climatic conditions. In addition, this study will be extended to more states in the country.

## 5. Conclusions

In this paper, the survival times of patients with COVID-19 in three states of the Mexican Republic were studied, taking into account the geographical locations of the cases, comorbidities, age, gender, and the development of pneumonia. Patient survival times differed by geographic location. Therefore, patient location is an important factor for COVID-19 survival times. The proportional hazard frailty model performed better than the Cox model. According to the results, the COVID-19 survival times of male patients were shorter compared with women; age also influenced the survival times of patients with COVID-19. Obesity, diabetes, CKD, and hypertension were comorbidities that increased the risk of death from COVID-19 in all three states. In the State of México, patients with asthma had a lower risk of dying than patients without asthma.

## Figures and Tables

**Figure 1 healthcare-12-00306-f001:**
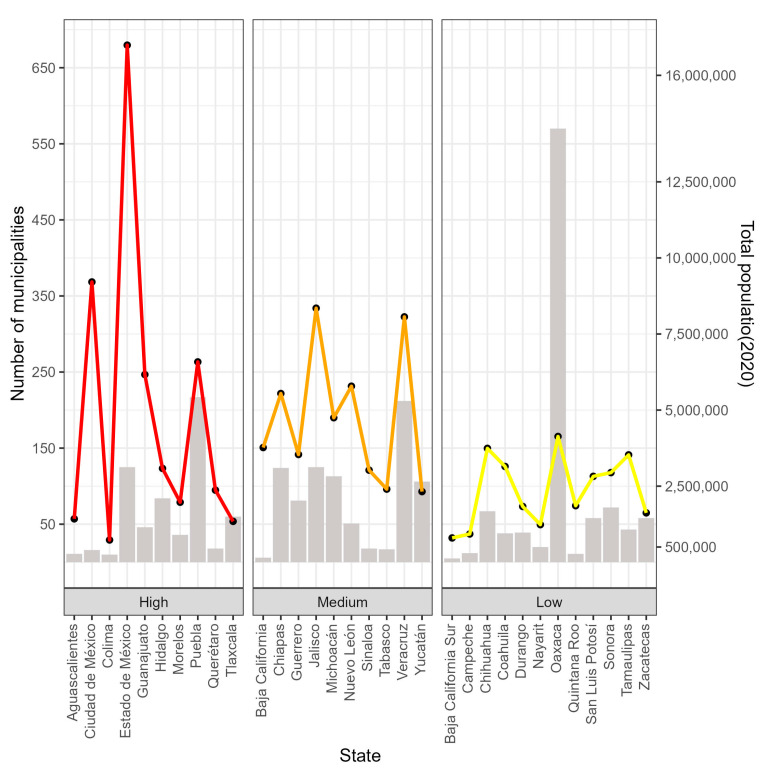
States of México with number of municipalities (bars and left axis) and total population (lines and right axis).

**Figure 2 healthcare-12-00306-f002:**
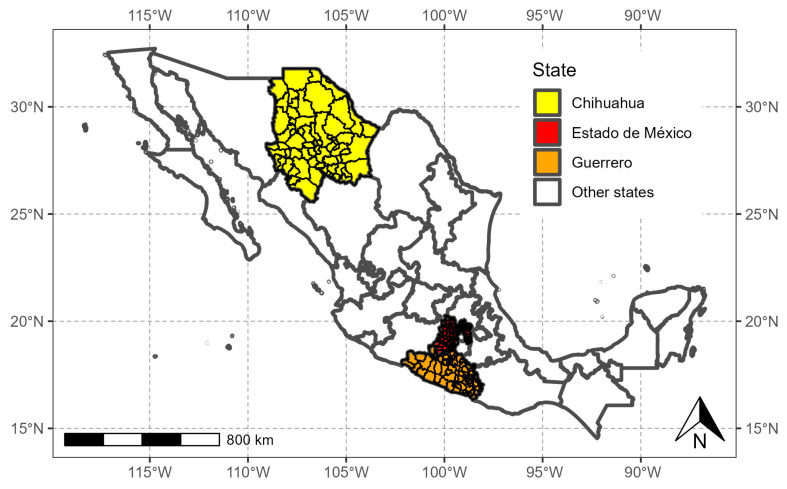
States of México selected for this study.

**Figure 3 healthcare-12-00306-f003:**
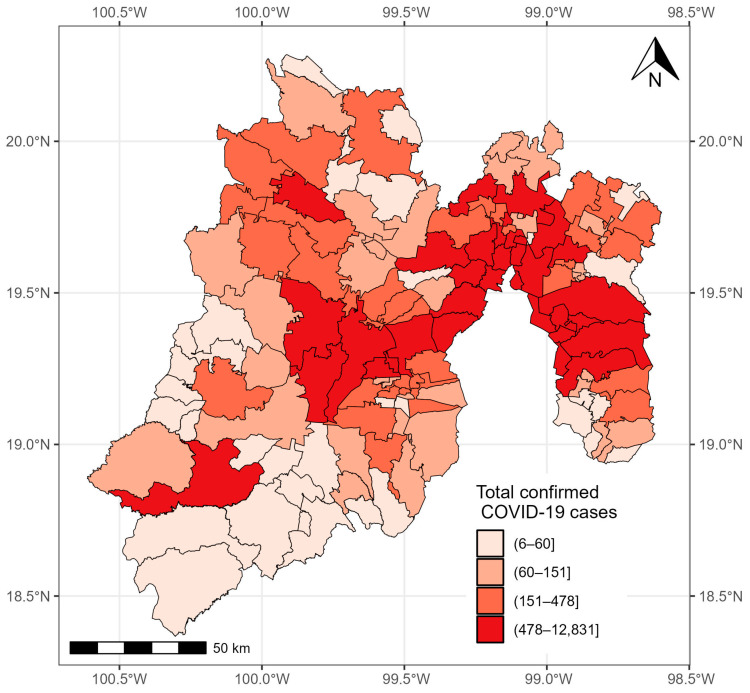
Confirmed cases of COVID-19 in the municipalities of the State of México from 28 February 2020 to 24 November 2021.

**Figure 4 healthcare-12-00306-f004:**
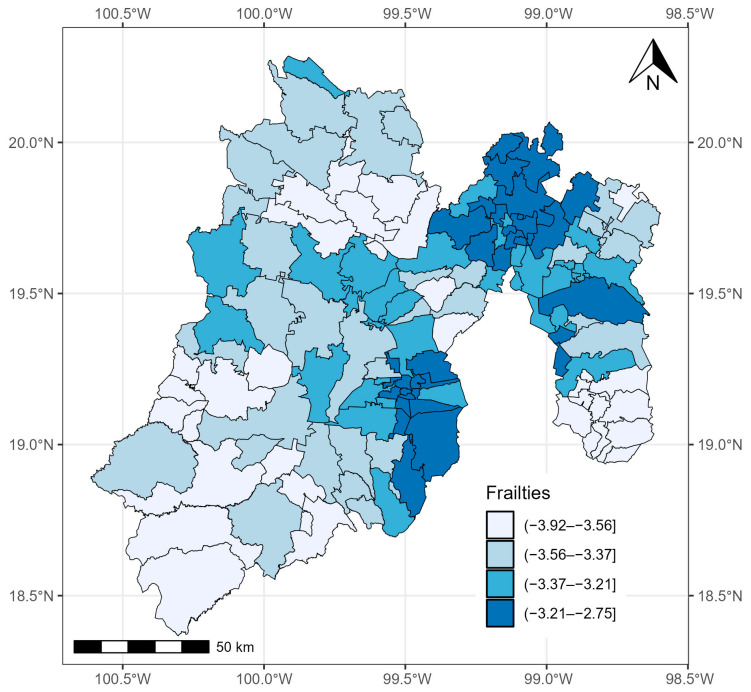
Quantiles of posterior samples for frailties in the State of México.

**Figure 5 healthcare-12-00306-f005:**
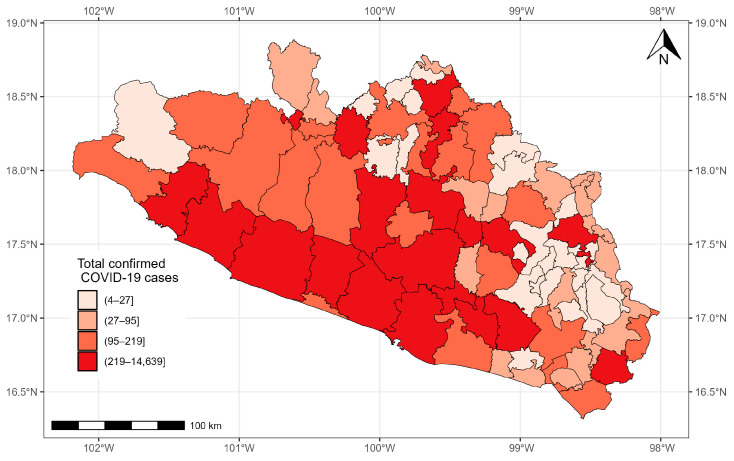
Confirmed cases of COVID-19 in the municipalities of Guerrero from 28 February 2020 to 24 November 2021.

**Figure 6 healthcare-12-00306-f006:**
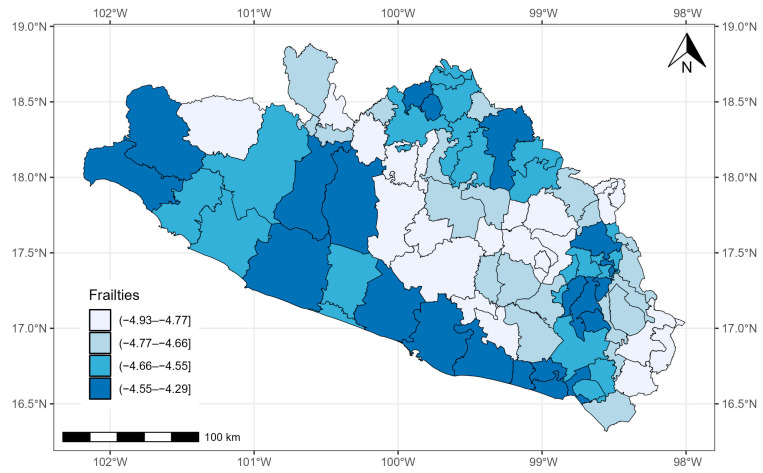
Quantiles of posterior samples for frailties in Guerrero.

**Figure 7 healthcare-12-00306-f007:**
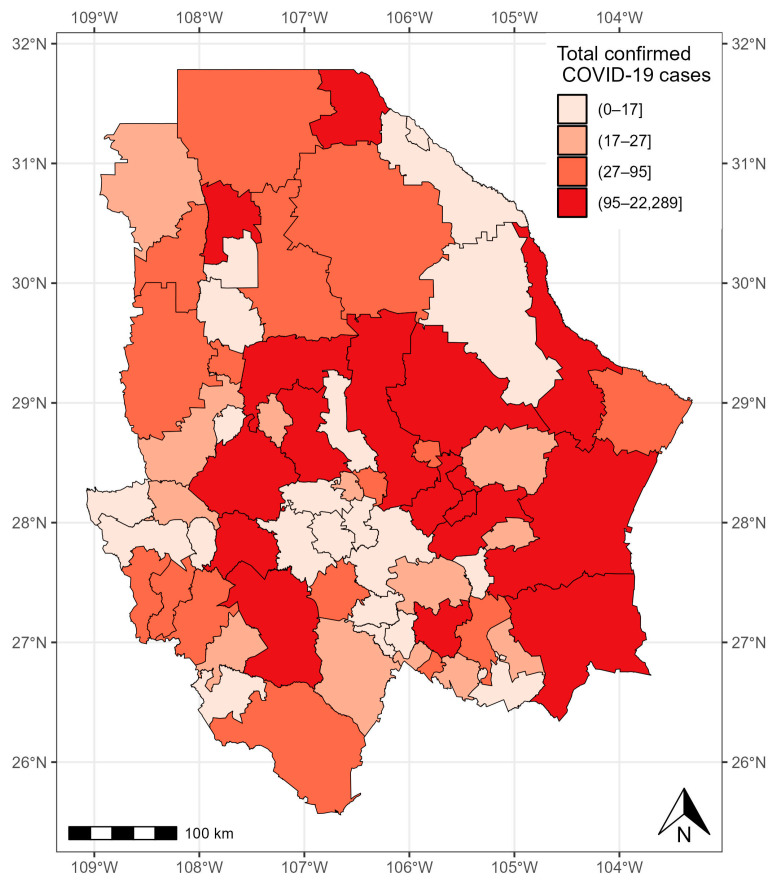
Confirmed cases of COVID-19 in the municipalities of Chihuahua from 28 February 2020 to 24 November 2021.

**Figure 8 healthcare-12-00306-f008:**
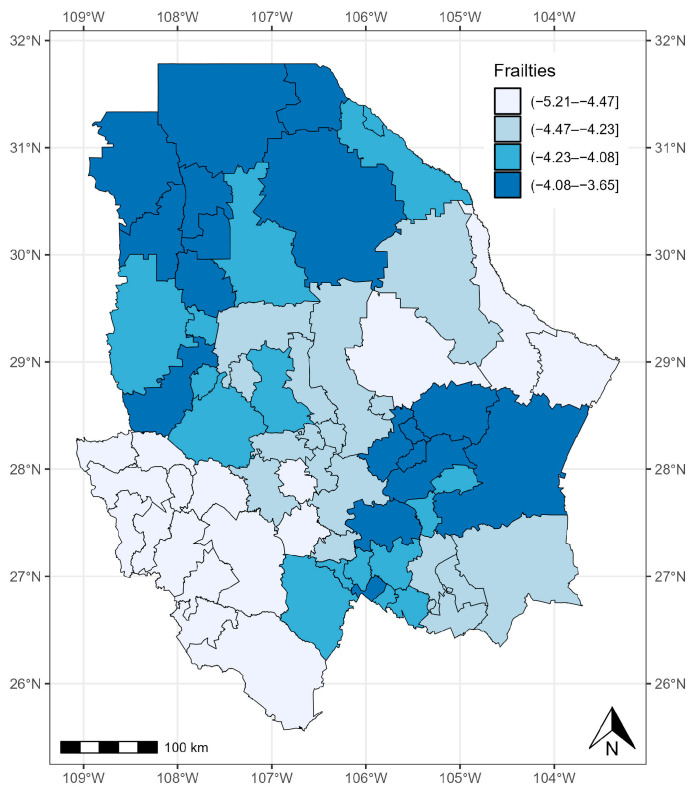
Quantiles of posterior samples for frailties in Chihuahua.

**Table 1 healthcare-12-00306-t001:** Variables considered in this study.

Variable	Code	Description
Age	Number of years	Patient’s age
Gender	1: Male; 2: Female	Identifies the gender of the patient
Pneumonia	1: Yes; 2: No	Indicates if the patient has pneumonia
Diabetes	1: Yes; 2: No	Indicates if the patient has diabetes
Chronic obstructive pulmonary disease	1: Yes; 2: No	Indicates if the patient has COPD
Cardiovascular disease	1: Yes; 2: No	Indicates if the patient has CVD
Obesity	1: Yes; 2: No	Indicates if the patient has obesity
Asthma	1: Yes; 2: No	Indicates if the patient has asthma
Chronic renal disease	1: Yes; 2: No	Indicates if the patient has CKD
Hypertension	1: Yes; 2: No	Indicates if the patient has hypertension

**Table 2 healthcare-12-00306-t002:** Estimated parameters of the spatial PH model for the State of México.

Variable	Mean	Median	95% CrI	$\hat{R}$	CrI-Upper
Age	0.042	0.042	(0.041, 0.043) **	1.01	1.02
Sex (male)	0.341	0.341	(0.308, 0.373) **	1.00	1.01
Pneumonia	1.680	1.679	(1.645, 1.719) **	1.00	1.02
Diabetes	0.164	0.164	(0.130, 0.197) **	1.01	1.03
COPD	0.017	0.017	(−0.066, 0.096)	1.00	1.00
CVD	−0.096	−0.094	(−0.186, −0.011) **	1.01	1.04
Obesity	0.129	0.129	(0.085, 0.170) **	1.01	1.07
Asthma	−0.228	−0.226	(−0.389, −0.081) **	1.02	1.08
CKD	0.463	0.463	(0.394, 0.535) **	1.01	1.05
Hypertension	0.063	0.064	(0.026, 0.098) **	1.00	1.00
$\hat{\alpha }$	0.068	0.066	(0.036, 0.110) **	1.66	2.97
${\hat{\tau }}^{2}$	1.551	1.464	(0.901, 2.692) **	1.01	1.05
$\hat{\phi }$	0.273	0.263	(0.133, 0.470) **	1.01	1.03
${\hat{\theta }}_{1}$	−4.167	−4.173	(−4.182, −4.131) **	-	-
${\hat{\theta }}_{2}$	−0.059	-0.061	(−0.071, −0.045) **	-	-

CrI: credible interval, $\hat{R}$: Gelman and Rubin’s convergence diagnostic, CrI-Upper: upper limit of the credible interval, ** 95% credible interval does not include zero.

**Table 3 healthcare-12-00306-t003:** Estimated parameters of the spatial PH model for the state of Guerrero.

Variable	Mean	Median	95% CrI	$\hat{R}$	CrI-Upper
Age	0.035	0.035	(0.033, 0.037) **	1.00	1.00
Sex (Male)	0.191	0.192	(0.127, 0.252) **	1.00	1.00
Pneumonia	2.819	2.818	(2.718, 2.913) **	1.00	1.01
Diabetes	0.304	0.305	(0.239, 0.368) **	1.00	1.00
COPD	0.140	0.140	(0.001, 0.277) **	1.00	1.00
CVD	0.206	0.206	(0.063, 0.354) **	1.00	1.00
Obesity	0.219	0.218	(0.146, 0.296) **	1.00	1.00
Asthma	−0.120	−0.119	(−0.343, 0.103)	1.00	1.02
CKD	0.196	0.197	(0.073, 0.325) **	1.01	1.03
Hypertension	0.070	0.071	(0.004, 0.132) **	1.00	1.00
$\hat{\alpha }$	0.076	0.072	(0.041, 0.126) **	1.03	1.11
${\hat{\tau }}^{2}$	1.838	1.731	(1.072, 3.166) **	1.00	1.00
$\hat{\phi }$	0.083	0.077	(0.032, 0.163) **	1.00	1.00
${\hat{\theta }}_{1}$	−4.315	−4.280	(−4.520, −4.198) **	-	-
${\hat{\theta }}_{2}$	0.174	0.174	(0.152, 0.208) **	-	-

CrI: Credible interval, $\hat{R}$: Gelman and Rubin’s convergence diagnostic, CrI-Upper: Upper limit of the credible interval, ** 95% credible interval does not include zero.

**Table 4 healthcare-12-00306-t004:** Estimated parameters of the spatial PH model for the state of Chihuahua.

Variable	Mean	Median	95% CrI	$\hat{R}$	CrI-Upper
Age	0.048	0.048	(0.046, 0.050) **	1.01	1.03
Sex (male)	0.310	0.310	(0.257, 0.365) **	1.00	1.00
Pneumonia	1.689	1.688	(1.630, 1.750) **	1.00	1.02
Diabetes	0.341	0.341	(0.283, 0.401) **	1.00	1.00
COPD	0.068	0.070	(−0.072, 0.209)	1.00	1.00
CVD	−0.021	−0.022	(−0.130, 0.087)	1.00	1.00
Obesity	0.277	0.277	(0.211, 0.342) **	1.00	1.00
Asthma	−0.141	−0.144	(−0.313, 0.029)	1.00	1.00
CKD	0.531	0.533	(0.420, 0.633) **	1.00	1.01
Hypertension	0.245	0.245	(0.183, 0.301) **	1.00	1.00
$\hat{\alpha }$	0.099	0.096	(0.051, 0.160) **	1.00	1.01
${\hat{\tau }}^{2}$	2.158	2.035	(1.261, 3.724) **	1.00	1.01
$\hat{\phi }$	0.099	0.093	(0.043, 0.185) **	1.00	1.01
${\hat{\theta }}_{1}$	−4.384	−4.362	(−4.593, −4.286) **		
${\hat{\theta }}_{2}$	0.155	0.159	(0.133, 0.178) **		

CrI: credible interval, $\hat{R}$: Gelman and Rubin’s convergence diagnostic, CrI-Upper: upper limit of the credible interval, ** 95% credible interval does not include zero.

**Table 5 healthcare-12-00306-t005:** DIC, WAIC, and LPML of models.

State	PH Model	DIC	WAIC	LPML
State of México	Spatial	186,805.2	186,809.3	−93,404.61
	Cox	193,904.9	194,385.3	−97,192.46
Guerrero	Spatial	46,866.87	46,871.20	−23,435.59
	Cox	48,396.29	48,395.89	−24,197.94
Chihuahua	Spatial	63,225.16	63,232.3	−31,616.2
	Cox	65,013.54	65,014.24	−32,507.11

## Data Availability

The analyzed data set can be consulted at the following link: https://www.gob.mx/salud/documentos/datos-abiertos-152127 (accessed on 25 November 2021).

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
