# Peer review of "Spatial Survival Model for COVID-19 in México"

_healthcare, 2024, doi:10.3390/healthcare12030306_

Round 1

Reviewer 1 Report

Comments and Suggestions for Authors

This study aims to investigate the risk factors associated with COVID-19 deaths in three Mexican states, performing spatial statistical analysis in order to take into account the spatial variability among the data. I think this is an interesting and up-to-date work and would be a good contribution to “International Journal of Environmental Research and Public Health”.

The Introduction is relevant and theory based.  The methods are appropriate and clearly presented, though the softwares used should be mentioned. Regarding the results, some details to the maps could be improved to ameliorate their presentation. In Discussion, the authors should focus on specific findings. Moreover, some structural changes are required. Please find attached detailed comments and suggestions.

Comments on the Quality of English Language

Moderate editing of English language required

Author Response

Muchas gracias por revisar el manuscrito. En el documento adjunto encontrará las respuestas a las observaciones sugeridas.

Reviewer 2 Report

Comments and Suggestions for Authors

Please find my comments attached herewith.

Comments on the Quality of English Language

The English is adequate.

Author Response

Thank you very much for reviewing the manuscript. In the attached document you will find the responses to the suggested observations.

Round 2

Reviewer 1 Report

Comments and Suggestions for Authors

I want to thank you for the time and effort you put in revising the manuscript, as well as for providing detailed responses to my comments and suggestions. I consider the revised manuscript as suitable for publication.

Comments on the Quality of English Language

Minor editing of English language required.

Author Response

Thank you for taking the time to review our manuscript again. We appreciate the valuable comments provided in this new revision. The suggestion for minor language editing was made. The manuscript was sent for proofreading to the services provided by MDPI (Certificate attached).

Reviewer 2 Report

Comments and Suggestions for Authors

I don't have any further comments except one suggestion. The authors provided the Cox-Snell residual plot to test the proportionality assumption. All the plots show the violation of the assumption. This is not new and quite common when it comes to Cox modeling. I would suggest the authors mention that (or better justify it) somewhere in the paper for completeness.

Author Response

Thank you for taking the time to review our manuscript again. We appreciate the valuable comments you provided in this new revision.
